# Non-Predictive Model-Free Control of Nonlinear Systems with Unknown Input Time Delay

**DOI:** 10.3390/e25071076

**Published:** 2023-07-17

**Authors:** Quanmin Zhu, Jianhua Zhang, Weicun Zhang

**Affiliations:** 1School of Engineering, University of the West of England, Coldharbour Lane, Bristol BS16 1QY, UK; 2School of Information and Control Engineering, Qingdao University of Technology, Qingdao 266525, China; jianhuazhang@qut.edu.cn; 3School of Automation and Electrical Engineering, University of Science and Technology Beijing, Beijing 100083, China; weicunzhang@ustb.edu.cn

**Keywords:** input delay systems, non-predictive and model-free control, low gain control, low complexity controller, stabilization

## Abstract

This study presents a general framework for the control of unknown dynamic systems with unknown input delay. A concise output feedback control system is structured with tuning stabilization/dynamic response by an output feedback low gain, removing steady state error against step reference with a feedforward gain. A series of stability analyses are presented for the designed control systems, (1) a gain/phase margin theorem is proposed for stability analysis by regulating the feedback gain, and (2) a stability theorem based on rational function approximation of the time delay is presented for dealing with the transcendental polynomial characteristic equations, which is equivalent to the analysis from the algebraic polynomial characteristic equation. Both approaches give coherent results for stability analysis by regulating the feedback gain. The approaches are applicable to nonlinear systems, which are linearizable in the neighborhood of the operating points. The low complexity of the controllers does not require hard analytical derivation/numerical calculations to produce an acceptable control performance for the considered systems. Several representative simulation case studies provide demonstrations of computational experiments against those analytically derived and guidance for potential applications.

## 1. Introduction

It has been well established that input delay is a critical challenge for the design of closed loop control in dynamic systems, and concise and effective solutions have value in academic research and applications [1,2] Mathematically, such systems are classified as functional differential equations (FDEs). They belong to infinite dimensional problems, when compared with ordinary differential equations (ODEs) [2]. There have been two major distinctive approaches for dealing with such delayed control system analysis and design. The most popular is the predictive control [3] and the second is non-predictive control [4,5]. For this focused study, it is not intended to make a comprehensive review of all the representative approaches over a great number of publications. To justify the motivation and contribution of the study, we select a few closely related representative methods for critical review.

The Smith prediction control [6] and the other generalized prediction control approaches [7] are effective, while plant model and delay are known to be accurate. On the other hand, they are very much less robust against uncertainties that are widely and frequently encountered in manmade systems (industrial processes and plants). Another problem is the computational complexity in the prediction of output/state by solving convolution integral to nonlinear systems [1]. Stabilization of linear time-delay systems has been studied extensively [8] in areas such as the approaches using finite spectrum assignment [9] and continuous pole placement [10,11]. However, stabilization of nonlinear time-delay systems is more challenging in analysis and design.

In contrast to model-based predictive control, non-predictive control approaches are, in the main, a class of control system designs that do not use a model to predict the time delay induced outputs and states; most of the studies are state space model based [4,5]. When considering a representative approach for designing a state feedback low gain vector [4], nonsingular transformation matrices need to be set up and a state vector is assumed to be available for applications. Output feedback stabilization, requires a plant model for the conversion [4]. Even output feedback from a polynomial model is still commonly used in most of the industrial operations and comparatively the scale of the studies is much smaller than that of the state space model-based approaches. This could be because of the difficulty in dealing with nonlinear polynomials in comparison to state space equations. It should be noted that it is much more convenient/straightforward dealing with the time delay in the frequency domain for linear systems, in which the time delay just induces a phase shift in the underlying system.

Regarding using variable structure control to deal with input time delay, the Sliding Mode Control (SMC) has been the most popular method adopted for designing the systems [12,13]. Most of the approaches assume linear models and small/known time delay. It should be noted that if the time delay does not appear in the input, then the other delay control system designs are simpler. Regarding using Active Disturbance Rejection Control (ADRC) to deal with input delay systems [14], there have been some fundamental works presented, such as Delayed Designed ADRC (DD-ADRC), Polynomial based Predictive ADRC (PP-ADRC), and Smith Predictor based ADRC (SP-ADRC) and Predictor Observer based ADRC (PO-ADRC) [15]. The approaches are well studied based on some conditions: (1) nominal models, (2) approximation of time delay with rational polynomials, and (3) small time delay and/or state prediction. With these results, some further development could be conducted in reducing complexity in analysis and design via a non-predictive model-free control. Dealing with nonlinear systems with input delay is still a generally open issue with ADRC methodology, for which linear approximation in the neighborhood of the operating points could be considered.

A model-free control (MFC) [16] has not been widely used to deal with input delay systems. With the author’s best efforts, only a few related publications were found [17,18]. The study of controls with unknown time delays is in its early stages, considering simple linear systems with good performances [17], which is based on an online adaptation of the non-physical design parameters of an ultra-local model. Further interesting research has been undertaken on the active disturbance rejection control for uncertain time-delay nonlinear systems [19], which is based on complicated mathematical functional analysis. It is noted that the Smith prediction plus model-free control has been integrated in order to configure a scheme for network control with time varying delay [18]. In general, model-free control of the systems with input delay is more challenging as there is less model knowledge available.

Regarding the other alleged model-free control approaches, the most popular one is the model-free adaptive control (MFAC) method, which has been widely studied in academic research and used in industrial applications. The basic idea of the MFAC is to establish a dynamic linear model of a nonlinear system at the current operating point (pointwise linearization). This is achieved using input/output data from the controlled system to estimate a set of pseudo partial derivatives online. A recently published representative example is Ardupilot-based adaptive autopilot: an architecture and software-in-the-loop experiment, a good reflection of the MFAC applications [20]. The MFAC has certainly been widely applied in dealing with the input time delay in the control systems [21]. It should be noted that the difference between the MFAC and the model-free control used in the study is whether a plant model is used, or not, as reference for a control system design. The MFAC still uses a data-driven online model in control system design. The other type of model-free control [22,23], which is derived from the Lyapunov positive definite stability criterion, does not use either an offline plant model or an online estimated plant approximation model in the control system design. The newly proposed model-free approach is a supplement to the MFAC.

One more point that needs to be clarified is that the study takes the linear approximation approach in dealing with nonlinear plant dynamics with input delay in the neighborhood of the operating points. Note that the control of “unknown dynamic systems with unknown input delay” is intrinsically nonlinear, whereas concepts like “gain/phase margin”, “algebra polynomial characteristic equation”, and so on, are based on linear systems theory, which is inevitably in an approximation to nonlinearities.

In brief, the above explained approaches and many others not quoted here have laid down solid foundations for further development in the direction of simple/effective non-predictive model-free approaches. This study deals with the stabilization of nonlinear systems subject to an unknown input delay, contrarily to the usual method from the literature, following the paradigm of non-predictive and model-free control, which can have high potential for applications. Accordingly, the main contributions of the study are justified below.

(1)The approach, model-free robust control of nonlinear dynamic systems with input delay, is general and concise for analytical expansions, simulation demonstrations, and applications.(2)The stability analysis for stabilizing the systems has established a platform for positively considering input delay into control system design. It can be even used for intentionally introducing delay into control system design (Jin, Niculescu, Boussaada, and Gu 2017).(3)The approach requests less knowledge of plants and merely assumes that the open loop is stable, output controllable, and closed loop stabilizable. There is no need for model structures/parameters and input delay time in the control system design, which, by trial-and-error, tunes the delay regulator output feedback gain $\alpha_{fb}$ within a narrow numerical range of $0<\alpha_{fb}<1$, within which the values have role in reducing the time delay effect in control system stabilization. The approach can increase the system gain stability margin $1/\alpha_{fb}$ times and phase stability margin to a $\left|\left|P_{G}(j\omega_{g})\right|-\left|P_{fb}(j\omega_{fbg})\right|\right|$ degree. Obviously, it is robust against uncertainties (the whole plant plus delay is treated as an uncertain black box with measurable output and enabled input at the two ends).(4)It provides a simulation portfolio for demonstrating the design procedure and explaining the transparent application procedure.(5)The study has application potential/significance in terms of providing simplicity (solutions) while dealing with unknown input-delayed dynamic complexity (problems).

The rest of the study is organized into four sections. Section 2 presents the preliminary research problem statement and the foundation techniques for the following development. Section 3 designs the control system and proposes two approaches analyzing/stabilizing the designed systems by regulating an output feedback low gain. Section 4 selects a wide range of representative examples, from pure delay, linear, and nonlinear plants, to simulate and analyze in order to validate the derived analytical results. Section 5 presents the conclusions of the study.

## 2. Preliminaries

### 2.1. Problem Formulation—Control of the Dynamic Plants with Delayed Input

Consider a class of general single input single output (SISO) nth order nonlinear dynamic plants, described by
(1)P:y(n)(t)=fy(0∼n−1)(t),u(t−h),
where y∈ℝ and u∈ℝ are the plant output and input, respectively, and y(0∼n−1)=y⋯y(n−1)T∈ℝn is the n−1 order output derivative vector. u(t−h) denotes that the input is h time delayed, and h∈ℝ+ is unknown.

Regarding control of plant (1), for which the control system configuration in block diagrams is shown in Figure 1, for the given output y(t−h) and reference operating signal r(t), the control input is generally formulated with u(t)=u(y(t−1),r(t))=Gcαffr(t)−αfby(t−h),αff∈ℝ,αfb∈(0,1)), where αff is the feedforward gain, αfb is the feedback gain, and Gc is the error correction controller.

### 2.2. Assumptions

**Assumption** **1.**f:u→y*, an unknown mapping from the input space to the output space is a continuously differentiable of class* ℂn *and satisfied with the Lipschitz continuity* f(x1)−f(x2)≤Kx1−x2, K∈ℝ+.

**Assumption** **2.***The plant is Bounded-Input-Bounded-Output (BIBO),* u(t)≤Bu∩y(t)≤By,∀t∈ℝ+,Bu,By∈ℝ+.

**Assumption** **3.***There exists an open neighborhood at the operating point, so that plant (1) can be approximated by a linear dynamic model with input delay at the operating points, which is expressed in the form of a Laplace transform* $Y(s)=e^{-sh}G(s)U(s)$.

**Assumption** **4.***The time delayed plant is stabilizable in a closed loop control. Accordingly, the upbound nature of the time delay* h *is varied with the stabilization condition of each underlying system.*

### 2.3. Linear Control Systems with Input Delay

This study uses a general SISO linear closed loop output feedback control system as a foundation to analyze the stability of the nonlinear control system shown in Figure 1, in the neighborhood of each operating point of the plant (1) without referring to the plant model and the time delay. Figure 2 shows the linear equivalent control system block diagram.

From Figure 2, the closed loop transfer function between output y∈ℝ and input u∈ℝ is expressed by,
(2)$$\frac{Y}{R}=\alpha_{ff}\frac{G_{c}G_{p}e^{-hs}}{1+\alpha_{fb}G_{c}G_{p}e^{-hs}}=\alpha_{ff}\frac{G_{cN}G_{pN}e^{-hs}}{G_{cD}G_{pD}+\alpha_{fb}G_{cN}G_{pN}e^{-hs}},$$
where Y and R are the Laplace transforms of the output and the input, respectively, $G_{c}$ and $G_{p}$ (a strictly proper rational function) are the Laplace transforms of the controller and the plant, respectively, and $e^{-hs}$ is the Laplace transform of the pure time delay h>0.

To facilitate the follow-up analysis, let
(3)$$\begin{array}{cc} G_{c}=\frac{G_{cN}}{G_{cD}} & G_{p}=\frac{G_{pN}}{G_{pD}}, \end{array}$$
where subscripts N and D denote the transfer function numerator and denominator polynomials, respectively.

Accordingly, the closed loop system characteristic equation, in terms of polynomials, is expressed as
(4)$$G_{cD}G_{pD}+\alpha_{fb}G_{cN}G_{pN}e^{-hs}=0.$$
This characteristic equation determines the underlying system stability with the location of its roots/zeros.

### 2.4. Routh–Hurwitz Stability Criterion (Patil 2021)

This study uses the Routh–Hurwitz stability criterion to convert the stability issues in the neighborhood of the operating point to a linear polynomial from a transcendental polynomial (due to the input time delay). The Routh–Hurwitz stability criterion [24] is a necessary and sufficient condition for analyzing the stability of linear time invariant (LTI) dynamical systems. The class of stable systems is bounded by its all-dynamic states as time goes to infinity. In formulation, consider an LTI dynamic system without time delay below,
(5)$$\frac{Y(s)}{R(s)}=G(s)=\frac{G_{N}(s)}{G_{D}(s)}=\frac{\beta_{0}s^{m}+\beta_{1}s^{m-1}+\cdots +\beta_{n-1}s+\beta_{m}}{\alpha_{0}s^{n}+\alpha_{1}s^{n-1}+\cdots +\alpha_{n-1}s+\alpha_{n}}\;n\geq m,$$
where *α*’s and *β*’s are constants. The stability of the system is determined by the roots (also called poles in control system analysis) in the denominator polynomial. Without factorization of the polynomial for the roots, there are two equivalent algorithms, the Routh test and Hurwitz matrix test, which just use the denominator polynomial coefficients to examine if all the poles with negative real parts are stable, that is, Re(polei)<0,i=1,⋯n, if it is a Hurwitz polynomial.

### 2.5. Poles of Control System with Input Time Delay

This study acknowledges the linear system stability issues induced by the input delay, which is rooted in the transcendent characteristic equation, and provides reference to deal with unstable poles on the right half of the S plane. Inspecting the characteristic Equation (4) in a linear closed loop control system, a transcendental equation has an infinite number of poles (roots). However, the number of those poles to the right half of the S plane is finite [11]. A continuous pole placement method has been developed to remove the identified unstable poles into left half of the S plane by tuning a small controller gain, while monitoring the other stable poles with large real parts. This approach is a model-based trial-and-error computational procedure.

### 2.6. Non-Predictive Low Gain Control

This study acknowledges the non-predictive low gain control for linear systems and points out an open space for further expansion. For describing a general class of continuous time SISO dynamic systems with input delay, consider a popular state space model of (Liu and Fridman 2014)
(6)x˙(t)=Ax(t)+Bu(t−τ(t)), x(0)=x0y(t)=Cx(t),
where the variables are the state vector x(t)∈ℝn, control input u(t)∈ℝ, u(t)=0, ∀t<0, time varying, and delay τ(t)∈0,h. This study lets τ(t)=h an unknown bounded constant, and output y(t)∈ℝ. For the feasibility of control and measurement, the stabilizable/controllable pair A,B and the observable pair A,C are assumed (Lin and Han 2007). For the corresponding stabilizing state feedback control, assign u(t)=Kx(t) to make the underlying system exponentially stable, and K is low gain vector. Accordingly, the control system is formatted with,
(7)x˙(t)=Ax(t)+Bu(t−h)=Ax(t)+BKx(t−h).

**Proposition** **1.***It has been proven that for any given arbitrarily large time delay* h≥0*, there exists the low gain vector* K *to make the closed loop system asymptotically stable [4]. It should be noted that even though the low gain vector is selected by trial-and-error, it is still required of the corresponding model that it design the output feedback law. This is a motivation for the study to develop a model-free approach for the total unknown plant (unknown dynamic structure and unknown time delay) with a concise output feedback control system configuration and narrow controller parameter tuning range.*

## 3. Stabilization of Dynamic Systems with Input Delay

### 3.1. Configuration of Linear Equivalent Control Systems

This study proposes a new control system configuration as shown in Figure 1, in which the feedback gain is $\alpha_{fb}$, the feedforward gain is $\alpha_{ff}$, and the cascade controller is $G_{c}$. The closed loop transfer function is expressed in (2), written down again for easy reference,
(8)$$\frac{Y}{R}=\alpha_{ff}\frac{G_{c}G_{p}e^{-hs}}{1+\alpha_{fb}G_{c}G_{p}e^{-hs}}=\alpha_{ff}\frac{G_{cN}G_{pN}e^{-hs}}{G_{cD}G_{pD}+\alpha_{fb}G_{cN}G_{pN}e^{-hs}}.$$The controllers are specified by (1) using the feedback gain $\alpha_{fb}$ to stabilize the system, (2) using the feedforward gain $\alpha_{ff}$ to remove steady errors in response to a step reference input, and (3) using the cascade controller $G_{c}$ in conjunction with $\alpha_{fb}$ and $\alpha_{ff}$ to improve the dynamic/steady state response.

**Remark** **1.***This is a type of two-degree freedom control [25], but it is the simplest one because (1) both stabilizing controller* $\alpha_{fb}$ *and setpoint tracking controller* $\alpha_{ff}$ *are constant gains instead of dynamic compensators, and (2) the designs are model-free instead of model (nominal model)-based, and are therefore more robust against uncertainty and disturbance.*

### 3.2. Controller Design

For the control system design, the study assumes (1) the time delay h is an unknown bounded constant, and (2) the closed loop is stabilizable. Therefore, the delay boundary varies from the underlying stabilized control systems.

For step reference input (a type of operating point), the objectives of designing the controllers with the configured control system in Figure 1 are explained in the following steps.

(1)Use the feedback gain $\alpha_{fb}$ to stabilize the system. It is not analytically calculated as there is no plant model available and the design follows a trial-and-error approach to select a proper $\alpha_{fb}$ in the range of $0<\alpha_{fb}<1$. The smaller of the $\alpha_{fb}$, the more of a stability margin, which is possibly a monotonic response. Increasing $\alpha_{fb}$ could generate a decayed oscillatory response, further increasing will drive the system to instability.(2)Use the feedforward gain $\alpha_{ff}$ to remove steady errors in response to a step reference input. This is achieved by the final value theorem referring to Figure 2. First assign $\alpha_{ff}=1$ it has,

(9)$$\underset{t\to \infty }{\lim}\;y(t)=\underset{s\to 0}{\lim}\;sY(s)=r{\left.\frac{G_{c}G_{p}}{1+\alpha_{fb}G_{c}G_{p}}\right|}_{s=0}.$$
Consequently, remove the steady error, let y(∞)=r give,
(10)$$\alpha_{ff}={\left.\frac{1+\alpha_{fb}G_{c}G_{p}}{G_{c}G_{p}}\right|}_{s=0}$$If controller $G_{c}$ has an integrator ($\frac{1}{s}$), the feedforward gain $\alpha_{ff}={\left.\frac{1+\alpha_{fb}G_{c}G_{p}}{G_{c}G_{p}}\right|}_{s=0}=\alpha_{fb}$.

For model-free control systems, by observing the steady state output response, select $\alpha_{ff}=\frac{r}{y(\infty )\in \alpha_{ff}=1}$.

(3)Use the cascade controller $G_{c}$ in conjunction with $\alpha_{fb}$ to improve the system dynamic/static performance. This is left for future studies and this study sets $G_{c}=1$.

### 3.3. Stability Analysis

This study takes Assumptions 1 to 4 to hold, presenting two approaches to conducting stability analysis from different angles.

#### 3.3.1. Open-Loop-Based Stability Analysis (Frequency Response/Gain and Phase Margins)

Now consider the closed-loop stability by analyzing the open-loop gain and phase margins in the frequency domain.

**Remark** **2.***In a closed loop control, stability analysis is the same at any point of the loop. Accordingly, for gain/phase margin analysis, it is feasible to take* $\alpha_{fb}G_{c}G_{p}e^{-hs}$ *. In order to focus on the role of* $\alpha_{fb}$ *for stability analysis, take* ${\left.\alpha_{fb}G_{p}e^{-hs}\right|}_{G_{c}=1}$.

The gain margin $g_{m}$ and phase margin $p_{m}$ are defined below [26]
(11)$$\begin{array}{l} g_{m}={\left|G(j\omega_{p})\right|}^{-1},∠G(j\omega_{p})=-180^{{}^{\circ}}\\ p_{m}=P(j\omega_{g})+180^{{}^{\circ}},\left|G(j\omega_{g})\right|=1 \end{array}.$$
Accordingly, the safe margins (before instability) are defined below.

In decibel
(12)$$\begin{array}{ll} g_{m}=-20\mathrm{lg}\left|G(j\omega_{p})\right|>0, & ∠G(j\omega_{p})=-180^{{}^{\circ}}\\ p_{m}=P(j\omega_{g})+180^{{}^{\circ}}>0, & \left|G(j\omega_{g})\right|=1 \end{array},$$
where lg is the logarithmic function with base 10.

In ratio
(13)$$\begin{array}{l} g_{m}={\left|G(j\omega_{p})\right|}^{-1}>1,\;∠G(j\omega_{p})=-180^{{}^{\circ}}\\ p_{m}=P(j\omega_{g})+180^{{}^{\circ}}>0,\;\left|G(j\omega_{g})\right|=1 \end{array},$$
let $G=G_{p}e^{-hs}$ refer to definition (11), let $\omega_{fbp}$ for the frequency having $∠\alpha_{fb}G(j\omega_{fbp})=-180^{{}^{\circ}}$, and $\omega_{fbg}$ for the frequency having $\left|\alpha_{fb}G(j\omega_{fbg})\right|=1$, then the regulated gain and phase margins of $\alpha_{fb}G_{p}e^{-hs}=\alpha_{fb}G$ are formulated as.

In decibel,
(14)$$\begin{array}{l} g_{fbm}=0-20\mathrm{lg}\left|\alpha_{fb}\right|-20\mathrm{lg}{\left|G(j\omega_{fbp})\right|}^{-1},\;∠\alpha_{fb}G(j\omega_{fbp})=∠G(j\omega )=-180^{{}^{\circ}}\\ p_{fbm}=P(j\omega_{fbg})+180^{{}^{\circ}},\;\left|\alpha_{fb}G(j\omega_{fbg})\right|=1 \end{array},$$
and in ratio,
(15)$$\begin{array}{l} g_{fbm}={\left|\alpha_{fb}G(j\omega_{fbp})\right|}^{-1},\;∠\alpha_{fb}G(j\omega_{fbp})=∠G(j\omega )=-180^{{}^{\circ}}\\ p_{fbm}=P(j\omega_{fbg})+180^{{}^{\circ}},\;\left|\alpha_{fb}G(j\omega_{fbg})\right|=1 \end{array}.$$

**Theorem** **1.***The feedback gain* $\alpha_{fb}$*, in a range of* $0<\alpha_{fb}<1$*, can regulate the margins to increase* $1/\alpha_{fb}$ *times in the gain margin and* $\left|\left|P_{G}(j\omega_{g})\right|-\left|P_{fb}(j\omega_{fbg})\right|\right|$ *degree in the phase margin for the corresponding closed-loop system.*

**Proof.** Assume the transfer function G(s) is properly rational and stable. Consider gain margin stability conditions in decibel satisfied below,

(16)$$g_{fbm}(dB):0-20\mathrm{lg}\alpha_{fb}-20\mathrm{lg}\left|G(j\omega_{p})\right|>0,\;∠\alpha_{fb}G(j\omega_{fbp})=∠G(j\omega )=-180^{{}^{\circ}}.$$
This can be derived as below,


(17)
$$g_{fbm}(dB):0-20\mathrm{lg}\alpha_{fb}-20\mathrm{lg}\left|G(j\omega_{p})\right|=20{\left|\mathrm{lg}\alpha_{fb}\right|}_{0<\alpha_{fb}<1}-20\mathrm{lg}\left|G(j\omega_{p})\right|>0.$$


Therefore, the gain margin increases $20\left|\mathrm{lg}\alpha_{fb}\right|$ dB. Convert it into a ratio expressing gain margin and this correspondingly increases $\frac{1}{\alpha_{fb}}$ times.

Consider the phase margin stability conditions satisfied below,
(18)$$p_{fbm}:P_{fb}(j\omega_{fbg})+180^{{}^{\circ}}>0,\;\left|\alpha_{fb}G(j\omega_{fbg})\right|=1.$$

As $\omega_{g}>\omega_{fbg}$, it gives $\left|p_{G}(j\omega_{g})\right|>\left|p_{fb}(j\omega_{fbg})\right|$. Consequently, the phase margin increases $\left|\left|p_{G}(j\omega_{g})\right|-\left|p_{fb}(j\omega_{fbg})\right|\right|$ degree. □

**Remark** **3.**
*The theorem is applicable to those nonlinear systems, satisfying Jacobian and Hessian conditions in the neighborhood of the operating point, which can be linearized.*


#### 3.3.2. Closed-Loop-Based Stability Analysis (Laplace Transfer Function/Routh Criterion)

Now consider the closed loop stability analysis of the input delay control system of (8), the characteristic equation is expressed by
(19)$$G_{cD}G_{pD}+\alpha_{fb}G_{cN}G_{pN}e^{-hs}=0.$$Let the denominator and numerator polynomials expressed below,
(20)$$\left\{\begin{array}{l} G_{cD}G_{pD}=s^{n}+\alpha_{1}s^{n-1}+\cdots +\alpha_{n-1}s+\alpha_{n}\\ G_{cN}G_{pN}=\beta_{1}s^{m}+\cdots +\beta_{m-1}s+\beta_{m} \end{array}\;m=n-1,\right.$$
and have both stable poles and zeros, that is, they represent the stable and minimum phase systems, which have been assumed as the necessity of stable open loop in some delay control system designs [15].

Add the delay factor $e^{-hs}$ to expand the characteristic Equation (19) in form of
(21)$$G_{cD}G_{pD}+\alpha_{fb}G_{cN}G_{pN}e^{-hs}=s^{n}+(\alpha_{1}+\beta_{1}\alpha_{fb}e^{-hs})s^{n-1}+\cdots +(\alpha_{n-1}+\beta_{n-1}\alpha_{fb}e^{-hs})s+(\alpha_{n}+\beta_{n}\alpha_{fb}e^{-hs})=0,$$
which is a quasi-polynomial where the s powers are multiplied by an exponential function as the delay factor [27].

**Theorem** **2.***With the first order rational approximation to the time delay factor* $e^{-hs}$*, the stability analysis has the following equivalency,*(22)$$G_{cD}G_{pD}+\alpha_{fb}G_{cN}G_{pN}e^{-hs}\overset{stability}{↔}\sum_{i=0}^{n+1}c_{i}s^{n+1-i}=c_{0}s^{n+1}+c_{1}s^{n}+\cdots +c_{n-1}s+c_{n},$$*where* $c_{i},i\in 0\cdots n$ *are the coefficients of the polynomial.*

**Proof.** Consider the first degree rational approximation (the same function in either the Taylor approximation or Pade approximation) of the time delay below [28],


(23)
$$e^{-hs}=\frac{e^{-hs/2}}{e^{hs/2}}\approx R_{1,1}(hs)=\frac{2-hs}{2+hs}.$$


The first order Pade approximation, like the other higher order approximations, has the role of introducing an unstable zero in open loop and, consequently, could induce unstable poles in the closed loop systems, which is the goal of the study: to express unstable zeros/poles introduced by time delay.

Substitute the approximation (23) into the transcendental characteristic Equation (21) and it gives,
(24)$$\begin{array}{l} G_{cD}G_{pD}+\alpha_{fb}G_{cN}G_{pN}e^{-hs}\overset{R_{11}}{\to }G_{cD}G_{pD}\left(2+hs\right)+\alpha_{fb}G_{cN}G_{pN}\left(2-hs\right)\\ =c_{0}s^{n+1}+c_{1}s^{n}+\cdots +c_{n-1}s+c_{n}=0 \end{array},$$
where $C(c_{0}\cdots c_{n+1})\in \left(\alpha ,\beta ,2,\pm h,\alpha_{fb}\right),\;\alpha (\alpha_{1}\cdots \alpha_{n}),\;\beta (\beta_{1}\cdots \beta_{m})$.

Therefore, the classical Routh array and Hurwitz matrix tests are applicable for delayed system stability analysis.

In general, the Pade approximation of order $\left(m_{p},n_{p}\right)$ to $e^{-hs}$ [28] is a rational function $P_{m_{p},n_{p}}(h,s)$ formulated in expression of
(25)$$P_{m_{p},n_{p}}(h,s)=\frac{q_{0}+q_{1}(hs)+\cdots +q_{m_{p}}{\left(hs\right)}^{m_{p}}}{p_{0}+p_{1}(hs)+\cdots +p_{n_{p}}{\left(hs\right)}^{n_{p}}},m_{p}\leq n_{p},$$
where
(26)qi=(−1)i(mp+np−i)!mp!(mp+np)!(mp−i)!i!,i∈[0,1,⋯mp]∈ℝpj=(mp+np−j)!np!(mp+np)!(np−j)!j!,j∈[0,1,⋯np]∈ℝ.

Accordingly, the nth order Pade approximation of the characteristic equation with the time delay factor $e^{-hs}$ (19) is expressed in forms of
(27)$$\begin{array}{l} G_{cD}G_{pD}+\alpha_{fb}G_{cN}G_{pN}e^{-hs}\overset{Pade(m_{p},n_{p})}{\to }G_{cD}G_{pD}+\alpha_{fb}G_{cN}G_{pN}P_{m,n}\\ =c_{0}s^{n+n_{p}}+c_{1}s^{n}+\cdots +c_{n-1}s+c_{n+n_{p}}=0 \end{array},$$
where $C(c_{0}\cdots c_{n+n_{p}})\in \left(\alpha ,\beta ,Q,P,\alpha_{fb}\right),\;\alpha (\alpha_{1}\cdots \alpha_{n}),\;\beta (\beta_{1}\cdots \beta_{m})$ and $Q=[q_{0}\cdots q_{m_{p}}],P=[p_{0}\cdots p_{n_{p}}]$.

Therefore, the classical Routh array and Hurwitz matrix tests are applicable for delayed system stability analysis. □

**Remark** **4.**
*This confirms the fact that both the transcendental polynomial (11) and algebra polynomial (13) could have finite poles on the Right Half Plane (RHP) in an S coordinate system (nb. a system is unstable as long as one unstable pole appears), even though the former has infinite stable poles [11].*


**Remark** **5.**
*It has been established that such delay systems have ‘a finite dimensional flavor’. There has been a remark [29] that the problems in delay systems, related to stability and optimality in the framework of hyperstability (or dissipativity/passivity called today), may be accommodated in the systems without delay [30].*


**Remark** **6.***The equivalency is true and becomes equality for the corresponding linear discrete time systems (Z transfer function modelled) with integer time delay (*$h\in Z^{+}$*). That is,*(28)$$G_{cD}G_{pD}+\alpha_{fb}G_{cN}G_{pN}z^{-h}=\sum_{i=0}^{N=n+h}c_{i}z^{N-i}+=z^{n+h}+c_{1}z^{n+j-1}+\cdots +c_{n+h-1}z+c_{n+h},\;(c_{0}=1),$$*where* $c_{i}\in f(\alpha_{fb}),\;i\in Z^{+}=[0,n+h]$.

**Remark** **7.***The differences, comparing the two characteristic equations in (21), are (1) one is a transcendental polynomial equation with infinite poles and the other is an algebra polynomial equation with finite poles (but it can have infinite groups of N poles through varying* $\alpha_{fb}$*), and (2) by tuning the feedback gain* $\alpha_{fb}$*, one is model-based (requesting the delay time known as well) including a trial-and-error iterative computation process [11] and the other is model-free (no need to know the delay time as well) trial-and-error iterative simulation process within a narrow parameter of tuning bands* $0<\alpha_{fb}<1$*, which is the kernel of the study.*

**Remark** **8.** ***(feasibility of nonlinear systems):***
*Assume a nonlinear plant can be approximated by a series of piecewise linearized models (e.g., transfer functions) [31] or pointwise linearized models [32] in the neighborhood of the operation points. Then, the above analyses are applicable to such nonlinear plants.*


**Remark** **9.** ***(a summary of the two stability analyses):***
*In principle, input time delay can be treated as a virtual gain, proportionally, and gain increases with delay increasing. It is a common understanding that the gain increasing can reduce the stability margin or cause instability in closed loop control of the dynamic systems. The delay cannot be changed, however, changing the feedback gain can equivalently reduce the delay. Therefore, reduce the virtual gain to increase the stability margin.*


## 4. Examples

This section selects several bench test plants of (1) to simulate to test the proposed control performance to a step reference input. It is noted that the plant models and time delay are just used for simulation rather than being used for the controller design, which represents the model-free control prototype. The Simulink simulations follow the general control system configuration as shown in Figure 1. For testing the examples’ performance in the computational experiments, set up,

(1)The operating point is assigned as a step reference with amplitude four for all of the examples.(2)Time delay h=5 for all the examples, and is only used for simulation, not for controller design, which means that the time delay is unknown.(3)The controller $G_{c}=1$ for all of the examples.(4)Using a trial-and-error iterative approach to determine the feedback gain $\alpha_{fb}$ case by case, which starts from the value 0.5 in the middle of the feedback controller boundaries (0,1) and then tries the two directions (increase/decrease) until a desired value is determined. Once $\alpha_{fb}$ is selected, calculate the feedforward gain $\alpha_{ff}$. It should be noted that this simulation study uses a unique choice $\alpha_{fb}=0.1$ for all the simulated control systems for comparison.

### 4.1. Pure Time Delay $G_{p}=e^{-hs}$

Two groups of controllers are used for simulation,

(1)$\begin{array}{cc} \alpha_{fb}=1 & \alpha_{ff}=1 \end{array}$, for full delay feedback;(2)$\begin{array}{cc} \alpha_{fb}=0.1 & \alpha_{ff}=4 \end{array}/3.636$, for reduced delay feedback.

Figure 3 illustrates both case simulations for comparison. To explain the output responses in both cases, consider the plant discrete time model (Z transform). The time delay is correspondingly converted as $G_{p}(s)=e^{-hs}\Leftrightarrow G_{p}(z)=z^{-h}(h\in Z^{+})$. Consequently, the closed loop transfer function is given by $\frac{Y(z)}{R(z)}=\frac{z^{-h}}{1+\alpha_{fb}z^{-h}}=\frac{1}{z^{h}+\alpha_{fb}}$. For $\alpha_{bf}=1$, the system pole is on the unit circle on *Z* plane to give such a constant output oscillation. For $0<\alpha_{fb}<1$, the system poles are within the unit circle to give such a decayed output response.

### 4.2. First Order Linear Plant with Input Delay $G_{p}=\frac{2e^{-hs}}{s+1}$

Two groups of controllers are used for simulation,

(1)$\begin{array}{cc} \alpha_{fb}=1 & \alpha_{ff}=1 \end{array}$, for full delay feedback;(2)$\begin{array}{cc} \alpha_{fb}=0.1 & \alpha_{ff}=4 \end{array}/0.9756$, for reduced delay feedback.

Figure 4 illustrates both case simulations for comparison. It shows clearly that the full delay feedback produces an unstable output response. By reducing the feedback gain to regulating the delay feedback effect, the output response is stabilized. An interesting observation is that this first order system can have an oscillatory response with the inclusion of the delay, otherwise it must add at least one more additional pole though a controller. This gives a positive impression of delay (increasing the additional number of poles) introduced in the control system design/operation [10].

Furthermore, the simulation results can be explained analytically. Consider the first degree rational approximation (the same function in either the Taylor approximation or Pade approximation) of the time delay below,
(29)$$e^{-hs}=\frac{e^{-hs/2}}{e^{hs/2}}\approx R_{1,1}(hs)=\frac{2-hs}{2+hs},$$
where $R_{1,1}(hs)$ is for the first degree rational approximation to the time delay $e^{-hs}$.

Accordingly, the considered example has the closed loop transfer function of
(30)$$\frac{Y}{R}=\frac{G_{p}e^{-hs}}{1+\alpha_{fb}G_{p}e^{-hs}}.$$

The corresponding characteristic equation gives,
(31)$$1+\alpha_{fb}G_{p}e^{-hs}=hs^{2}+(2+h-2\alpha_{fb}h)s+2+4\alpha_{fb}=0.$$

For the general second order systems with characteristic equation $\alpha_{0}s+\alpha_{1}s+\alpha_{2}=0$, it is Hurwitz stable if all the coefficients are positive, that is $\alpha_{i}>0,\;i\in [0,2]$. Clearly the stability condition is $(2+h-2\alpha_{fb}h)>0$.

For the full feedback control $\alpha_{fb}=1$, the characteristic equation has
(32)$${\left[1+\alpha_{fb}G_{p}e^{-hs}\right]}_{\alpha_{fb}=1,h=5}=5s^{2}-3s+6=0.$$

Therefore, it is unstable. The oscillatory response is because of the complex conjugate poles in the characteristic equation.

Correspondingly the open loop gain margin and phase margin are,
(33)$$\begin{array}{cc} g_{m}={\left.0.35dB\right|}_{\omega_{cg}=0.825rad/s} & p_{m}={\left.-66.57\deg\right|}_{\omega_{cp}=3.856rad/s} \end{array}.$$

The negative phase margin indicates that the closed loop is unstable.

For reduced (low gain) feedback control $\alpha_{fb}=0.1$, the characteristic equation has,
(34)$${\left[1+\alpha_{fb}G_{p}e^{-hs}\right]}_{\alpha_{fb}=0.1,h=5}=5s^{2}+6s+2.4=0.$$

Therefore, it is stable. The oscillatory response is because of the complex conjugate zeros.

Correspondingly the open loop gain margin and phase margin are
(35)$$g_{m}={\left.3.5dB\right|}_{\omega_{cg}=0.825rad/s}\;p_{m}={\left.+\infty \right|}_{\omega_{cp}=NaN}.$$

Both margins are positive, therefore the closed loop is stable. Further comparing the gain margins in the two cases confirms the analytical expectation that the low gain control improves the gain margin $\frac{1}{\alpha_{fb}}=\frac{1}{0.1}=10$ times. Moreover, the example can represent a cease while the control of nonlinear plant (1) is in the neighborhood of its operating point, which is in the form of a linearized approximation.

### 4.3. Second Order Plant with Input Delay $G_{p}=\frac{2e^{-hs}}{s^{2}+s+1}$

Two groups of controllers are used for the simulation,

(1)$\begin{array}{cc} \alpha_{fb}=1 & \alpha_{ff}=1 \end{array}$, for full delay feedback;(2)$\begin{array}{cc} \alpha_{fb}=0.1 & \alpha_{ff}=4 \end{array}/6.664$, for reduced delay feedback.

Figure 5 illustrates both case simulations for comparison. It shows clearly that the full delay feedback produces an unstable output response. By reducing the feedback gain to regulating the delay feedback effect, the output response is stabilized.

Once again, the analytical derivations should be coherent with the simulated derivations, while following the routine explained with Section 4.2. Furthermore, the example can represent a cease while the control of nonlinear plant (1) is in the neighborhood of its operating point, which is in the form of a linearized approximation.

### 4.4. Van de Pol (VDP) Oscillator Dynamics with Input Delay

The Van der Pol oscillator is composed of a non-conservative system with nonlinear damping to follow a second-order dynamic [33]. For the VDP system with input delay, replacing u(t) with u(t−h) throughout the system gives,
(36)$$y^{\left(2\right)}(t)=\mu (1-y(t))y^{\left(1\right)}(t)-y(t)+u(t-h),$$
where μ=1.5 for the nonlinear damping strength.

Two groups of controllers are used for the simulation,

(1)$\begin{array}{cc} \alpha_{fb}=1 & \alpha_{ff}=1 \end{array}$, for full delay feedback;(2)$\begin{array}{cc} \alpha_{fb}=0.1 & \alpha_{ff}=4 \end{array}/3.636$, for reduced delay feedback.

Figure 6 illustrates both case simulations for comparison. This shows that both designed systems are stable, one with decayed oscillatory output response, the other with a monotonic output response. Clearly, increasing the decay regulation gain makes the system response decayed oscillatory, and constantly oscillatory ($\alpha_{fb}\geq 1.5$, tested, but not shown in the figures). Furthermore, the other tests indicate that the control of the delay system is stable with $\alpha_{fb}>0$.

### 4.5. Nonlinear Nonaffine Plant with Input Delay

The selected nonlinear nonaffine dynamic system [22] with input delay is expressed as
(37)$$\begin{array}{c} {\dot{x}}_{1}(t)=x_{2}(t)\\ {\dot{x}}_{2}(t)=-0.6x_{2}(t)-x_{1}(t)x_{2}(t)-u(t-h)x_{2}(t)+\sin(u(t-h))+2u(t-h)+u^{3}(t-h)\\ y(t)=x_{1}(t) \end{array}.$$

Two groups of controllers are used for the simulation,

(1)$\begin{array}{cc} \alpha_{fb}=1 & \alpha_{ff}=1 \end{array}$, for full delay feedback;(2)$\begin{array}{cc} \alpha_{fb}=0.1 & \alpha_{ff}=4 \end{array}/14.5$, for reduced delay feedback.

Figure 7 illustrates both case simulations for comparison. For $\alpha_{fb}=1$, the system is unstable because of the large gain effect. Obviously, by regulating the delay feedback gain (reducing $\alpha_{fb}$), the system is stabilized and a good decayed oscillatory output response profile is achieved. Again, further reducing $\alpha_{fb}$ could generate decayed monotonic output responses ($0<\alpha_{fb}\leq 0.05$ with simulated tests).

### 4.6. Nonlinear Rational Plant with Input Delay

A nonlinear rational model, a ratio of numerator differential polynomial against denominator differential polynomial, has appeared widely in chemical engineering processes [34] and has been used as bench test to control complex nonlinear dynamic systems [35,36]. The input delay is a critical challenge in such control system design. A bench test example is considered below,
(38)$$\begin{array}{c} {\dot{x}}_{1}(t)=x_{2}(t)\\ {\dot{x}}_{2}(t)=\frac{-4x_{1}(t)-x_{2}(t)+4u(t-h)}{1+0.1x_{1}^{2}(t)+0.1u^{2}(t-h)}\\ y(t)=x_{1}(t) \end{array}.$$

Two groups of controllers are used for the simulation,

(1)$\begin{array}{cc} \alpha_{bf}=1 & \alpha_{ff}=1 \end{array}$, for full delay feedback;(2)$\begin{array}{cc} \alpha_{bf}=0.1 & \alpha_{ff}=8/7.25 \end{array}$, for reduced delay feedback.

Figure 8 illustrates both case simulations for comparison. For $\alpha_{bf}=1$, the system is slow, oscillatory, and unstable (observed by expanding simulation time to 4000 because of the large gain effect). Obviously, by regulating the delay feedback gain (reducing $\alpha_{bf}$), the system is stabilized with decayed oscillatory output response. As the model is an open loop decayed oscillatory dynamic, from simulation observation, when further reducing $\alpha_{bf}$ until the feedback is disconnected ($\alpha_{bf}=0$) the system output responses move towards the open loop response. It is noted that such control system cannot achieve a monotonic response to a step reference, which could be further studied to regulate the system’s dynamic response and stabilization in the future.

Summary discussions of the simulation results

(1)The simulation tests demonstrate the analytical results derived from stability analysis and control system design.(2)The input delay increases the dynamic order of the characteristic equation, which easily drives open loop stable systems into instability with full delay feedback in the corresponding closed loop systems. It is observed that through reducing output feedback gain $\alpha_{fb}$ (regulating delay feedback), the systems can be comfortably stabilized for the closed loop stabilizable systems.(3)All tests show that unstable responses are oscillatory (instead of monotonic unstable). The oscillatory instability comes from the fact of at least one pair of complex conjugate poles have a positive real part, which is due to the exponential components in the system transfer function.(4)The approach is a model-free control for linear/nonlinear systems with unknown constant input delay. There is no need for prediction, non-physical design parameters of an ultra-local model, when compared with aforementioned popular model-free control approaches.(5)Regarding the decay regulation, tuning $\alpha_{fb}$ within a narrow trial-and-error range provides an easy iterative testing/tuning procedure (similar to engineering tuning on site–connect to tune). The selection of $0<\alpha_{fb}<1$ increases stability gain/phase margins, robustness, etc. Both limits of $\begin{array}{cc} \alpha_{fb}=0 & \alpha_{fb}=1 \end{array}$ represent the open loop operation and full decay feedback operation, respectively.(6)The simulations–computational experiments are mainly designed for validating feedback stabilization using the decay regulator $\alpha_{fb}$. For dealing with the steady error referring to a step reference, the feedforward gain $\alpha_{ff}$ has been determined by $\alpha_{ff}=\frac{r}{y(\infty )\in \alpha_{ff}=1}$ case by case. For a unilateral approach to remove such steady errors, a simulation was assigned to the cascade controller $G_{c}$ with a Proportional and Integral (PI) transfer function, which has been tested (not shown in the figures) with the nonlinear nonaffine dynamic system of (37). The results confirm the feedforward gain $\alpha_{ff}=\alpha_{fb}$, which is consistent with that derived in Section 3. Using integral control to remove the steady error against step/level reference has been the general procedure in analytical work and applications. As explained in Section 3, $G_{c}$ will be systematically studied in the future work.

## 5. Conclusions

This study proposes a concise approach, using output feedback low gain to stabilize control systems with unknown dynamic and input delay. The stability analysis is presented in both open loop (gain and phase margins) and closed loop (Routh/Hurwitz array/matrix). The feedback gain controller $0<\alpha_{fb}<1$ tuning is trial-and-error based, but within a very narrow range (0, 1), and similar to onsite tuning. Simulation tests are consistent with those analytically derived. This creates the potential to introduce input delay intentionally for designing future control systems within a concise framework. The other topic left out of the study is the analysis and design of the controller $G_{c}$ integrated with $\alpha_{fb}$ and $\alpha_{ff}$, which should provide a more flexible/enhanced control effect. Another potential study could be expanding the approach to control systems with variable time delay. Wide range bench tests are needed in order to provide comprehensive assurance for future applications.

It should be noted that, essentially, the role of the feedback low gain $\alpha_{fb}$ is to stabilize such systems (closed loop stabilizable input time delay systems; without $\alpha_{fb}$ the systems could easily be unstable). In the simulated examples, $\alpha_{fb}$ cannot be tuned to make the desired dynamic/transient response (such as that specified by the damping ratio and undamped natural frequency), with the exception of stabilization and the following dynamic response. This is a constraint of the current study. Hopefully in future studies the error correction controller $G_{c}$, as shown in Figure 1 and Figure 2, will play a role in transient response regulation.

## Figures and Tables

**Figure 1 entropy-25-01076-f001:**
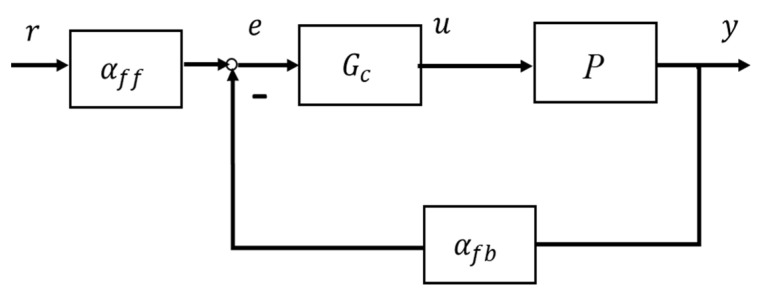
Nonlinear systems with unknown plant model.

**Figure 2 entropy-25-01076-f002:**
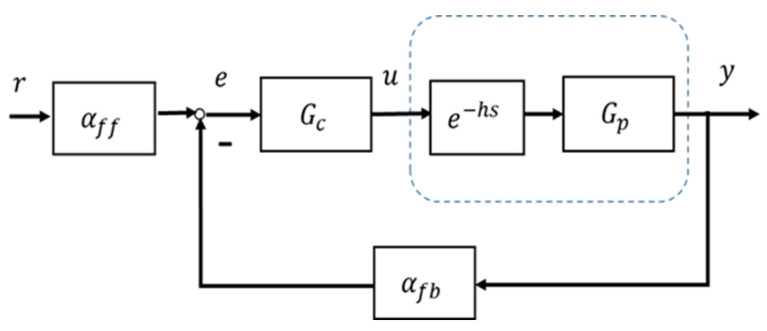
Linear equivalent control system with input time delay.

**Figure 3 entropy-25-01076-f003:**
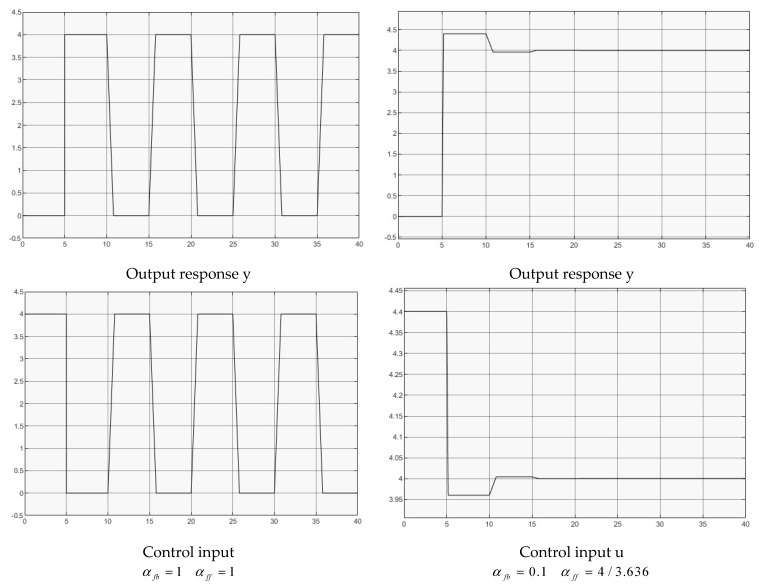
Control of pure time delay.

**Figure 4 entropy-25-01076-f004:**
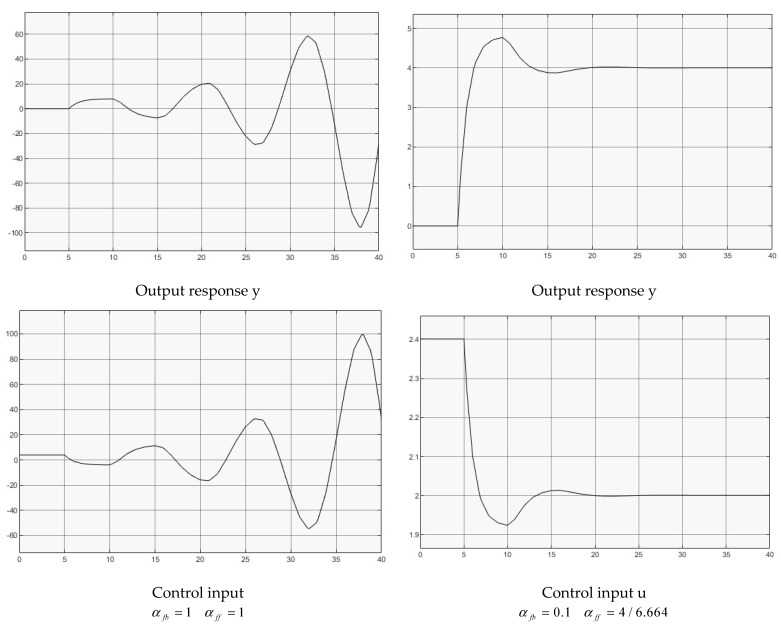
Control of first order transfer function.

**Figure 5 entropy-25-01076-f005:**
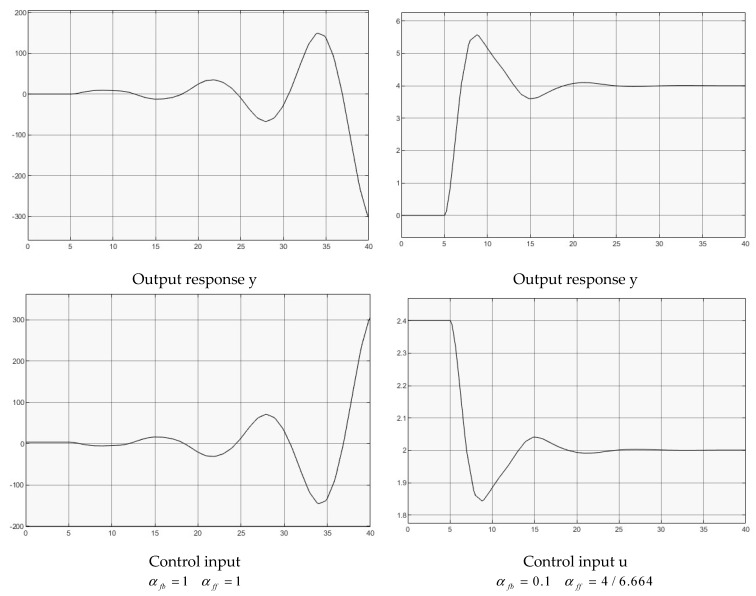
Control of second order transfer function.

**Figure 6 entropy-25-01076-f006:**
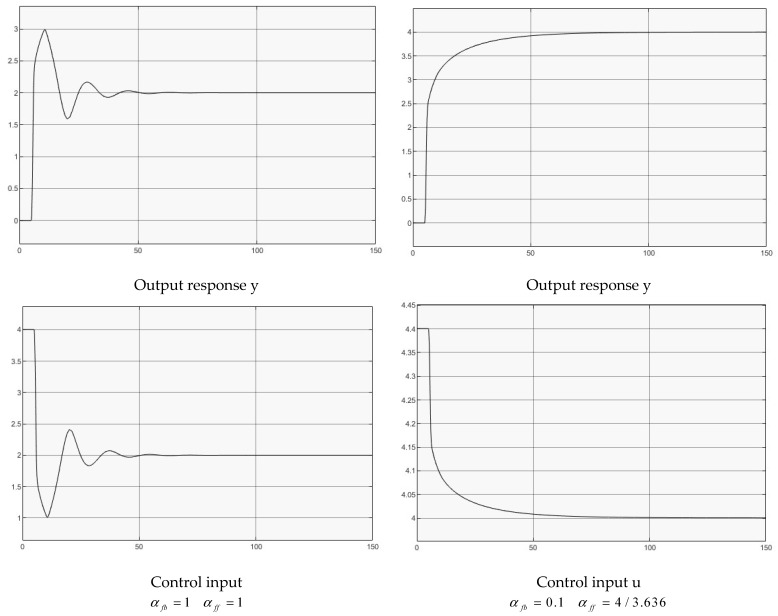
Control of VDP oscillator.

**Figure 7 entropy-25-01076-f007:**
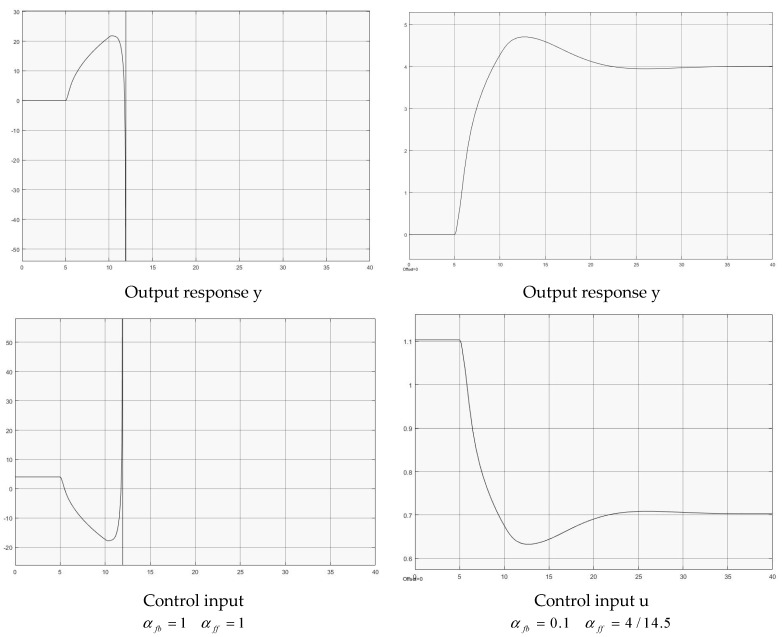
Control of nonlinear nonaffine plant.

**Figure 8 entropy-25-01076-f008:**
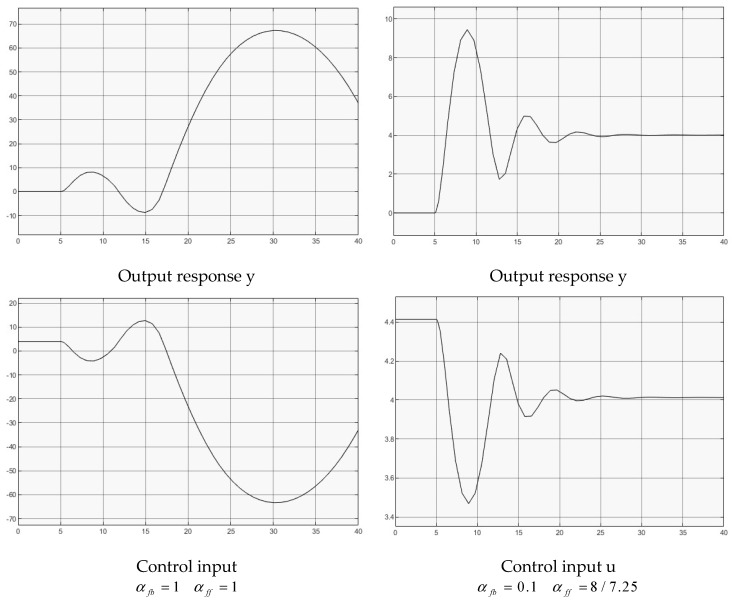
Control of nonlinear rational plant.

## Data Availability

The datasets generated during and/or analysed during the current study are available from the corresponding author on reasonable request.

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
