# Peer review of "Non-Predictive Model-Free Control of Nonlinear Systems with Unknown Input Time Delay"

_entropy, 2023, doi:10.3390/e25071076_

Round 1
Reviewer 1 Report
The author presents a general framework for the control of unknown dynamic systems with unknown input delay. First of all, an output feedback control system is structured with tuning stabilisation/dynamic response by an output feedback low gain and removing steady state error against step reference by a feedforward gain. Secondly, a series of stability analyses are presented for the designed control systems. Finally, several representative simulation case studies provide the demonstrations of computational experiments against those analytically derived and the guidance of the design procedures. The topic is interesting and the paper is well organized. The paper is acceptable on improving by the followings:
(1) The concept of ‘non-predictive’ should be clarified.
(2) In the part 4, examples section, more nonlinear system examples are welcome to show the effectiveness of the proposed approach.
Author Response
Please find the enclosed respond report. Thanks

Reviewer 2 Report
The aim of the research is significant, but the elaboration and the discussion should be improved. Some important issues are not clear.
In the introduction explain why is the range of alfajb chosen between 0 and 1? (line 108).
How is the trial and error method executed not considering the model of the plant? (line 205)
Is it enough to consider only the first order Pade (not Pada) approximation of the time delay? How big can be the time delay compared to the values of the other time constants in the system?
In the examples explain why do you choose the alfa values as you do? (lines 336, 347, 388, 404, 418).
In the nonlinear examples it is not clear how the linearization is executed?
In the references 9 and 10 belong to the same paper, 24 and 25 also.
English should be checked.
Author Response

(The authors gave the same response as above.)

Reviewer 3 Report
Attached

Author Response
Please see the attachment. Thank you for your help.

Reviewer 4 Report
This study presents a general framework for the control of unknown dynamic systems with unknown input delay. The topic is appropriate for the journal, hwever I find it strange that the authors are not aware of the topic of model-free adaptive control MFAC and of its recent applications: Ardupilot-based adaptive autopilot: architecture and software-in-the-loop experiments; On model-free adaptive control and its stability analysis. these suggestions could be relevant since, quoting "A concise output feedback control system is structured with tuning stabilisation/dynamic response by an output feedback low gain and removing steady state error against step reference by a feedforward gain" this philosophy is closely related to MFAC, although not the same. The authors are free to judge if the suggestions are appropriate. Concerning the theory, the authors do this
"
A series of stability analyses are presented for the designed control systems,
1) a gain/phase margin theorem is proposed for stability analysis by regulating the feedback gain,
2) a stability theorem based on rational function approximation of the time delay is presented for dealing with the transcendental polynomial characteristic equation, which is equivalent to the analysis from the algebra polynomial characteristic equation.
"
Although these analysis could be reasonable, the author fail to acknowledge that they are based on some appropriximation. Note that control of "unknown dynamic systems with unknown input delay" is intrinsically nonlinear, whereas concepts like "gain/phase margin" "algebra polynomial characteristic equation" are based on linear systems theory that is necessarily an approximation. I belive that if the authors present in a more clear way that their methods are linear approximations of the actual nonlinear situation, the work can be accepted. Currently, the manuscript contains too many sentences that overstate the contribution
- "Both approaches give the coherent results for stability analysis". Please notice that both approaches are based on approximations
- "The low complexity of the controllers does not require hard analytical derivation/numerical calculations". Of course the method is low complexity, but this is because the authors approximate a nonlinear situation with a linear one
- "The whole control system is analysed with crisp control concept/principles/formulations". the so-called crisp control concepts are based on linear approximation, can they really be called crisp?
In other words, the work can be accepted after revisions
This study presents a general framework for the control of unknown dynamic systems with unknown input delay. The topic is appropriate for the journal, hwever I find it strange that the authors are not aware of the topic of model-free adaptive control MFAC and of its recent applications: Ardupilot-based adaptive autopilot: architecture and software-in-the-loop experiments; On model-free adaptive control and its stability analysis. these suggestions could be relevant since, quoting "A concise output feedback control system is structured with tuning stabilisation/dynamic response by an output feedback low gain and removing steady state error against step reference by a feedforward gain" this philosophy is closely related to MFAC, although not the same. The authors are free to judge if the suggestions are appropriate. Concerning the theory, the authors do this
"
A series of stability analyses are presented for the designed control systems,
1) a gain/phase margin theorem is proposed for stability analysis by regulating the feedback gain,
2) a stability theorem based on rational function approximation of the time delay is presented for dealing with the transcendental polynomial characteristic equation, which is equivalent to the analysis from the algebra polynomial characteristic equation.
"
Although these analysis could be reasonable, the author fail to acknowledge that they are based on some appropriximation. Note that control of "unknown dynamic systems with unknown input delay" is intrinsically nonlinear, whereas concepts like "gain/phase margin" "algebra polynomial characteristic equation" are based on linear systems theory that is necessarily an approximation. I belive that if the authors present in a more clear way that their methods are linear approximations of the actual nonlinear situation, the work can be accepted. Currently, the manuscript contains too many sentences that overstate the contribution
- "Both approaches give the coherent results for stability analysis". Please notice that both approaches are based on approximations
- "The low complexity of the controllers does not require hard analytical derivation/numerical calculations". Of course the method is low complexity, but this is because the authors approximate a nonlinear situation with a linear one
- "The whole control system is analysed with crisp control concept/principles/formulations". the so-called crisp control concepts are based on linear approximation, can they really be called crisp?
In other words, the work can be accepted after revisions
Author Response

(The authors gave the same response as above.)

Round 2
Reviewer 2 Report
Please check English.
Explain more clearly why you did not use higher order Pade approximations for the time delay?
In example 4.6 the stabilization provides still a bit oscillating response. Can this behavior improved further?
Please check English.
Author Response
Thank you so much for your suggestions. We have revised your paper as you suggested.

Reviewer 3 Report
The authors have made significant improvements, the result has become much clearer and now the reviewer has no doubts about the expediency of publication. The reviewer acknowledges with satisfaction the inappropriateness of formulations about "tasks solved for 100 years", etc.
At the same time, the authors must admit that the first evaluation of the article was completely provoked by an extremely unsuccessful presentation of the result.
In the presented revision the main troubles have been eliminated. The only thing that remains unclear is the lack of a clearly formulated task. Yes, all the provisions of the statement and conditions can be found in the text. But in different places and with different connotations. Why the authors do not want to collect and systematize all the necessary words and, possibly, formulas in one strict paragraph of the "Problem statement" remains unclear. The reviewer strongly recommends making such a paragraph, section.
The second thing that the authors will have to do for publication is to arrange the text, especially the formulas, in accordance with the requirements of the journal.
Author Response
Thank you for your suggestions. Please find attached files.

Reviewer 4 Report
The paper has been submitted in a revised form with all comments addressed
Author Response
Thank you so much for your encouragement.